# Personalized Nutrition Approach in Pregnancy and Early Life to Tackle Childhood and Adult Non-Communicable Diseases

**DOI:** 10.3390/life11060467

**Published:** 2021-05-24

**Authors:** Shaikha Alabduljabbar, Sara Al Zaidan, Arun Prasath Lakshmanan, Annalisa Terranegra

**Affiliations:** Mother and Child Health Program, Research Department, Sidra Medicine, P.O. Box 26999, Doha, Qatar; salabduljabbar@sidra.org (S.A.); salzaidan@sidra.org (S.A.Z.); alakshmanan@sidra.org (A.P.L.)

**Keywords:** precision nutrition, pregnancy, breastfeeding, non-communicable diseases, gut microbiota, nutrigenetics, epigenetics, transcriptomics

## Abstract

The development of childhood and adult non-communicable diseases (NCD) is associated with environmental factors, starting from intrauterine life. A new theory finds the roots of epigenetic programming in parental gametogenesis, continuing during embryo development, fetal life, and finally in post-natal life. Maternal health status and poor nutrition are widely recognized as implications in the onset of childhood and adult diseases. Early nutrition, particularly breastfeeding, also plays a primary role in affecting the health status of an individual later in life. A poor maternal diet during pregnancy and lack of breastfeeding can cause a nutrient deficiency that affects the gut microbiota, and acts as a cofactor for many pathways, impacting the epigenetic controls and transcription of genes involved in the metabolism, angiogenesis, and other pathways, leading to NCDs in adult life. Both maternal and fetal genetic backgrounds also affect nutrient adsorption and functioning at the cellular level. This review discusses the most recent evidence on maternal nutrition and breastfeeding in the development of NCD, the potentiality of the omics technologies in uncovering the molecular mechanisms underlying it, with the future prospective of applying a personalized nutrition approach to prevent and treat NCD from the beginning of fetal life.

## 1. Introduction

Nutrition plays an important role at all life stages—before and during pregnancy, lactation, infancy, and childhood, as well as during adult life. Maternal nutrition has a major impact on the infant, not only because the nutrient exchange through the placenta and breast milk is involved in fetal and infant growth, but it also plays a role in determining the offspring’s risk of developing non-communicable diseases (NCDs) [1,2,3]. NCDs are defined as non-infectious diseases that progress slowly, but become chronic; they usually require long-term treatment. NCDs include cardiovascular disease (CVD), type 2 diabetes (T2D), metabolic syndrome, etc. [1]. Recent studies have investigated the roles of pregnancy and infancy as the most critical stages that influence the risks of NCDs in childhood and adult life. New nutritional recommendations were developed to reduce the burden of NCDs in future generations [4].

In the last few years, the food science have started to analyze the effects of nutrients and dietary behaviors on cellular functions and gene modulation This new approach is defined as precision nutrition, where genetic background as well as microbiota are taken into account, to understand the response to diet and to single nutrients intake, and to tackle metabolic diseases (e.g., NCDs) [5,6].

Since there is limited evidence for personalized nutrition in pregnancy and early life, this review aims to summarize current data and perspectives on the roles of diet, nutrigenetics, gut microbiome, epigenetics, and transcriptomics in pregnancy and breastfeeding, contributing to develop NCDs in the offspring.

## 2. Maternal Diet and NCDs

Maternal health status and poor nutrition are widely recognized as implications in the onset of childhood and adult diseases [7]. A poor maternal diet can cause a nutrient deficiency that, as mentioned before, acts as a cofactor for many pathways, impacting the control of genes involved in the metabolism, angiogenesis, cognitive development, and other pathways, leading to chronic diseases in adult life. The mechanism is known as the “window of susceptibility” to nutritional programming in the fetal life [8] and refers to vulnerability of the fetus to environmental factors, such as the maternal diet, affecting health outcomes determined by the timing of the exposures [9]. In the 1980s, the paradigm of the Developmental Origins of Health and Disease (DoHAD) was developed. Utilizing multiple studies from different geographical areas, researchers observed how famine affected the following generations [10]. The Dutch famine resulted in low caloric intake, below 1000 Kcal per day, for a certain period of time. Therefore, researchers investigated whether the maternal diet had a role in the offspring developing NCDs [11]. Researchers examined the long-term consequences of starvation on the fetuses after many years. The Dutch famine data showed that the prevalence of coronary artery disease was significantly higher in fetuses exposed to undernutrition compared to non-exposed fetuses [12]. Ravelli et al. found also that exposure to famine during fetal life resulted in the development of glucose intolerance in adulthood [13]. Moreover, adults who were exposed to the Chinese famine during gestational life showed significantly higher systolic and diastolic blood pressure, hypertension, and a higher risk of developing metabolic syndrome, compared to those who were not exposed to the famine [14]. Type 2 diabetes (T2D) was higher in offspring exposed to the Biafran famine [15] and the Ukraine famine, where the analysis showed that the more severe the exposure, the higher the risk [16]. According to prior studies on human and non-human models, authors showed an association between low maternal protein intake during pregnancy and an increase in the systolic blood pressure in the offspring later in life [17,18,19]. On the opposite side, a study on 569 children showed that infants of mothers who followed the Mediterranean diet during gestation period had lower blood pressure during childhood [20]. This finding is supported by another study that used a rat model fed with a high-fat diet during pregnancy; it showed an association with hypertension in the offspring [21]. Moreover, pre-conceptional obesity was found to be a risk factor, creating adverse metabolic effects in the mother and children [22,23]. An obesogenic diet during pregnancy after the fertilization stage can lead to the fetus acquiring morbidities, such as hyperinsulinemia and hypercholesterolemia, which are related to obesity, and increase the disease vulnerability throughout adult life [24].

Researchers are looking at potential mechanisms that explain the link between maternal nutritional status and offspring developing NCDs. The hypothesis is that adaptations occur in cases of nutrient deficiencies in fetuses, in order to give priority to the vital organs, such as the brain, and to maintain normal growth; therefore, the growth and functioning of the non-vital insulin-sensitive organs (e.g., the liver and pancreas) are compromised [25]. This can happen by blood flow redistribution to the organs, as well as reduction in secreting anabolic hormones, such as insulin, with reduced performance of these organs, which can continue later in life, causing disease [26]. Other theories suggest that the impact can start from the oocyte meiosis process until the baby’s birth. Intake of nutrients in the preconception period could affect the oocyte and later the fetal development, as referring to the recent research from Santangelo et al., who drew attention to the role of polyphenol intake in inhibiting the oocytes apoptosis and follicle atresia during ovarian developing stages [27].

## 3. Breastfeeding and NCDs

Many studies demonstrated the beneficial effects of breastfeeding in reducing the risk of NCDs. It has been shown that breastmilk plays a protective role against obesity [28,29,30,31] compared to formula milk [32]. This might be due to differences in the components in terms of nutrients and hormones of each type of milk. The protein content is higher in formula milk than in breast milk, while breast milk contains the leptin hormone that is lacking in the formula milk [33]. A study showed an association between the high fat and protein content in baby formula and the high secretion of insulin growth factor-type 1 (IGF-1), which consequently stimulated the adipocytes, resulting in weight gain [34]. According to in vitro studies, leptin that is present in the breast milk could possibly affect the growth factors and prevent the formation of the adipocytes [35,36]. Furthermore, breastfeeding has an impact on the protein and calorie intake [37], secretion of insulin [38], size of adipocyte, as well as maintaining balanced fat reserves [39]. Moreover, breastfeeding is suggested to prevent type 2 diabetes (T2D) in adulthood [40,41,42,43]. Different studies indicated that infants consuming formula milk have increased level of insulin compared to breast-fed infants [38,44,45], leading to a change in glucagon and insulin release, which play a role in early development of insulin resistance and T2D. This phenomenon can be explained again by the different compositions between the two types of milk, but also by the amount of formula milk intake that is usually much higher than breast milk [46]. Furthermore, breastfeeding is associated with a reduction of the main risk factors for cardiovascular diseases in adulthood [43,47], such as a high level of cholesterol [43,48], high blood pressure [43,48], and a low density lipoprotein (LDL) level [40]. In addition, high density lipoprotein (HDL) levels in adulthood can be improved by breastfeeding during infancy [49,50]. Finally, because of reduced sodium content in the breast milk, breast-fed infants are at a lower risk of developing hypertension during adult life [51].

These (and other) studies suggest the importance of nutrition during pregnancy and early life in determining the development of NCDs later in adulthood. The molecular mechanisms underlying these events are still far from being fully clarified. The approach of precision nutrition is to identify the molecular mechanisms underlying the effect of diet on the onset on NCD, and for this purpose, it uses the multi-omics approach, which is able to detect changes at a molecular level. The results from this multilevel analysis will elucidate the specific mechanisms of each nutrient on single tissues, cell types, and genes in a personalized medicine fashion.

## 4. Omics Technologies Applied to the Precision Nutrition in Pregnancy

The science of precision nutrition is relatively new and is taking advantages of all the recent technical advances in the omics technologies, such as genomics, transcriptomics, epigenomics, and microbiome. Precision nutrition merges the traditional diet and nutritional status analyses with multi-omics, enabling the capacity to define the individual response to diet and a personalized treatment for each single patient other than the “one-size-fits-all” approach traditionally used in diet therapy. In the following sections, we will discuss the current knowledge on single omics applications in pregnancy and offspring health status; in particular, we will focus on studies performed on the gut microbiota, nutrigenetics, epigenetics, and transcriptomics.

### 4.1. Diet and Gut Microbiota in NCD

The human microbiota refers to the community of 10–100 trillion microorganisms living in different sites of the human body, with a majority in the intestinal tract. The term microbiome refers to the genome of the microbiota [52]. Microbial dysbiosis has been shown to be related to some NCDs, such as obesity and autoimmune disorders [53,54,55,56,57,58]. That is because microbiota was found to be involved in host metabolism regulation, immunological responses, as well as in other physiological pathways [54,58,59,60]. Due to the fact that diet [61,62] and breast-feeding [61,63] are some of the environmental factors influencing the gut microbiota composition and abundance [60], maternal diet effects on offspring developing NCDs might be interfered by the microbiome.

#### 4.1.1. Maternal Diet and Gut Microbiota

The New Hampshire Birth Cohort Study data indicated that infant gut microbiota is influenced by maternal diet. The researchers showed that infant gut *Streptococcus* and *Clostridium neonatale* were positively and negatively correlated with maternal fish and seafood consumption, respectively. Moreover, a decrease in abundance of genus *Bifidobacterium* in the infants’ gut was related to increased consumption of fruit by the mothers during pregnancy; however, the authors suggested that the results can be influenced by the mode of the delivery as well [64]. Another study, conducted on primate models, confirmed the role of the maternal diet on the offspring’s microbiota alterations. Two groups of *Macaca fuscata* animal mothers were given different diets, one was fed with standard chow (CTD), consisting of 13% fat from soya bean oil, and the other group with a high fat diet (HFD), consisting of 36% fat from different sources. The offspring were delivered vaginally and breastfed until 6 to 7 months, when they either maintained their mothers’ diets or switched to the other diet, ending up with four offspring cohorts. This study data illustrated that persistent microbial dysbiosis was shown in the offspring of mothers fed with HFD during gestation, where Epsilonproteobacteria *Campylobacter* spp. and *Helicobacter* spp. were depleted and Firmicutes *Ruminococcus* and *Dialister* were enriched [65].

Since there is not yet a clear causal pathway of maternal diet during pregnancy affecting the health of the neonates, it was suggested that microbiome may have an influential role. Metabolic diseases, such as obesity and insulin resistance in children, were found to be associated with maternal obesity and dietary intake, correlating with infant gut microbiota alteration [66]. Hansen et al. examined the effect of feeding non-obese diabetic pregnant mice with gluten-free diets on their offspring. The co-authors found that the female offspring showed a decrease in gut bacterial levels of *Cyanobacteria* and *Deferribacteres,* as well as a reduction in diabetes incidence. After 4 and 10 weeks from birth, the offspring presented an increase in *Verrucomicrobia*, *Proteobacteria*, and *TM7* abundances and these bacteria were reported having a preventive role against diabetes development [67]. Similarly, a maternal low protein diet showed an alteration of the male offspring microbiota in early development in which consistent high *Roseburia intestinalis* levels were shown; this bacterial taxa was reported to be inversely correlated with atherosclerotic lesions [68], as well as its abundance, was low in T2D patients [55]. Another study conducted on mice described the effects of the maternal diet on the offspring’s intestinal dysbiosis as the pregnant dam mouse model fed with a western-style diet caused high abundance of *Clostridiales* and adverse outcomes, such as autoimmunity [69]. The mechanism of how the maternal diet alters the infant microbiome is still not clear. However, it is thought that the infant obtains the mother’s microbiome through swallowing the amniotic fluid [70] as some studies have demonstrated the similarity of bacterial taxonomies between the placenta and amniotic fluid, and the oral and meconium of the neonate [71,72,73].

#### 4.1.2. Breastfeeding Effect on Offspring Gut Microbiota

Human milk is another factor that impacts the development and maturation of the microbiome of the newborn, which is closely related to the mother’s milk and skin microbiome [74]. The breast milk prebiotic compounds, like the human milk oligosaccharides (HMO), are metabolized by the infant gut microbiome [75], and indorse bacterial community growth [76]. HMOs were found to a have a protective role in the development of obesity and autoimmune diseases in mouse models [77,78]. Based on Savage et al., the abundance of *Bifidobacteria*, *Lactobacillus*, and *Clostridia* were positively correlated with the duration of breastfeeding [79]; when breastfeeding was compared to formula feeding, it was shown to significantly impact the health of the infant by reducing the risk of developing obesity [80,81], with an increase in the breastfeeding duration [82]. Moreover, breastfeeding showed a protective role against type 1 diabetes acting on the gut microbiome [74,83].

Many of these studies have been conducted on non-human models, showing an association between diet during pregnancy, lactation period, and the microbiota alterations, leading to NCDs in their offspring (summarized in Figure 1). Further investigations are needed to validate these findings in human subjects.

### 4.2. Nutrigenetics Role in Offspring NCD

#### 4.2.1. Maternal Diet and Nutrigenetics of NCD

The interaction between the mother’s nutrition during pregnancy and her genetics background impacts the fetal nutrient source; hence, these alterations could influence fetal growth and development [84,85]. The nutritional genetics, which is called nutrigenetics, refers to a different response to nutrients and dietary factors due to gene variants [86]. We summarize here—below and in Figure 1—the most recent findings in nutrigenetics. An example of the interaction between genes and nutrients leading to a disease is the genetic variants of the methylenetetrahydrofolate reductase (*MTHFR*) gene with a single nucleotide polymorphism (SNP), in which a cytosine is substituted with a thymine, resulting in the amino acid change [87]. This SNP reduces the enzyme efficiency causing the accumulation of homocysteine in the plasma and, as a consequence, reducing the bioavailability of folate and vitamin B12 [88]. These two latter nutrients play a very important role in cell proliferation and differentiation, thus low availability or intake of folate and vitamin B12 by pregnant women carrying this SNP could affect the embryogenesis [89]. In addition, disturbances of maternal folate and vitamin B12 levels during pregnancy may contribute to the development of T2D and adiposity after 6 years of age, as stated by Yajnik et al. [90]. Maternal over-nutrition, leading to diabetes and obesity, may affect the function of the placenta and the fetal metabolism by nutrients provided through the placenta. Changes in these nutrients possibly provoke the proadipogenesis effect, as well as hyperinsulinemia in the fetus [91]. High levels of lipids in pregnant women could lead to accumulation of lipids in the placenta [92], and the fetal supply of crucial, long chain fatty acids are affected [93]; thus, impacting the growth and development of the fetus [94]. A genome-wide association study(GWAS) that used the largest number of individuals to identify maternal and fetal variants contributing to gestational weight gain (GWG) found some evidence that maternal SNPs may be a factor of GWG more than fetal SNPs. These preliminary data propose that the relation between GWG and the later outcome of the offspring possibly reflect the role of intrauterine environment [95]. A study on 950 African mother–child pairs, aimed to explore whether the genetic predisposition to adulthood obesity and birthweight was related to the mother’s obesity and whether an unbalanced diet could be a contributing factor. The study found that fetuses who were genetically susceptible to obesity showed a significant negative correlation with birthweight when the mother was also at a high genetic risk of obesity [96]. Changes in birthweight were reported to be linked to nutrients availability in the intrauterine environment [97]. In addition to unhealthy diet intake during pregnancy, polymorphisms in the mother and infant genes were shown to affect fetal health negatively [98]. For instance, Fatty Acid Desaturase 1 (*FADS1*) and Fatty Acid Desaturase 2 (*FADS2*) genes encode delta-5 and delta-6 desaturase enzymes, which are involved in the metabolism of polyunsaturated fatty acids (PUFA) [99]. Genetic variations in these genes, in mothers, was associated with a decrease in eicosapentaenoic acid (EPA) and arachidonic acid (AA) levels in the breast milk as well as in the baby blood [100,101]. Abnormal levels of maternal omega-3 and omega-6 fatty acids were reported to be correlated with low birth weight (LBW) [102], and this outcome was linked with CVD [103]. Moreover, abnormal activity of these enzymes affects the glucose metabolism, leading to a direct relation with diabetes risk [104]. Research was conducted on newborns affected by maturity-onset diabetes of the young (MODY) to study the association between birthweight and inherited mutations in the glucokinase gene (*GCK*) that causes a reduction in the pancreatic beta cells sensing to the glucose molecules. The study discovered that the baby birthweight was higher than the average when the mother carried the *GCK*–MODY gene mutation, because of the high maternal blood glucose level that activated high insulin secretion. The birthweight was reduced in case the fetus carried the mutation. So, it was suggested that alteration in the birthweight reflected the fetal insulin secretion induced by the fetal genotype as well as by the maternal hyperglycemia and her genotype [105]. Other studies, taking into account the birthweight, were conducted with scope to analyze the correlation between T2D risk loci variants, influencing the secretion of the insulin, and the birthweight. These studies concluded that 2 fetal risk alleles at the CDK5 regulatory subunit-associated protein 1-like 1 gene (*CDKAL1*) and 2 alleles at hematopoietically expressed homeobox/insulin-degrading enzyme (*HHEX/IDE*) loci were related to a lower birthweight. Interestingly, fetuses who carried the four risk alleles and had reduced birthweights were the ones whose mothers smoked three cigarettes daily in the last three months of pregnancy [106].

#### 4.2.2. Breastfeeding and Nutrigenetics of NCD

Early nutrition could also contribute to the health of newborns. Human breast milk (HBM) shows a protective influence from developing metabolic disorders later in life even in individuals who are genetically susceptible to these diseases [107]. The peroxisome proliferator-activated receptor-γ (PPARγ2) is a transcription factor that is expressed in fat cells and regulates insulin sensitivity. A study found that adolescents who were not breastfed in their early lives and carried the polymorphism PPARγ2 Pro12Ala, had increased BMI, waist circumference, and skinfold thickness compared to the ones who consumed breast milk, regardless of the breastfeeding duration [107]. Similarly, variations in the PPARγ2 gene were linked to a high risk of developing obesity in adulthood [108]. Since this gene ligand is present in the HBM, it was suggested to reduce the transcriptional activity of PPARγ2 in individuals with the polymorphism [109]. Moreover, PPARγ2 deficiency was found to contribute to increased levels of lipid oxidation enzymes in the lactating mammary gland and high levels of oxidized free fatty acids were observed in the offspring of mothers lacking this gene [110].

The impact of the nutrients on the interaction between metabolism and genetic variability is complex and this can explain the scarcity of studies on the genome and diet interaction in pregnant women and the infant susceptibility to NCD. Further research will give a better view on how and what nutrients taken during the gestation period can be affected by the maternal and fetal genetic background, resulting in a disease status in the children, hence the protection from those diseases. As a “one-size fits-all” approach is not recommended to be applied in the precision medicine revolution, then nutritional recommendations will be more valid if they are based on the patient’s nutrigenetics background.

### 4.3. Epigenomics Role in Offspring NCD

#### 4.3.1. Maternal Diet and Epigenetics of NCD

It has been proven that unhealthy diet contributes to NCDs and health problems [111,112,113]. However, the mechanisms are still not completely understood and seem to involve also the epigenetic modifications, which by definition affect the gene expression without changing the DNA sequence. Epigenetic changes can be established during fetal development and known to get impacted by the environment including maternal nutrition [114] leading to changes in the phenotype, such as NCD [85] (Figure 1). Diet can influence the epigenetics via different mechanisms, including histones modifications, DNA methylation and microRNA (miRNA). Histones modifications occur in the proteins that are bounded with DNA base pairs (histones) by covalent addition or removal of functional groups that alter the chromatin structure, hence affecting the gene expression [115]. DNA methylation is the epigenetic mechanism that transfers a methyl group onto specific cytosine residues of the DNA chain, modifying the DNA helix structure and affecting the binding of the transcription complex to DNA transcription site [116]. The third mechanism of epigenetics is the miRNA regulation. The miRNAs are 18–25 nucleotides non-coding RNAs that modulate the gene expression by binding to the untranslated regions of the mRNA in order to repress the translation of proteins and deteriorate the mRNA [117].

Nutrients may change the structure of chromatin, as lysine and arginine found in the histone N-terminal tails [118]. Moreover, many nutrients work as methyl-donors (e.g., methionine, choline, folate), directly providing the substrates for the methylation reactions; other nutrients, such as vitamin B2, B6, and B12, are cofactors of the enzymes involved in the methylation and, finally, nutrients (e.g., polyphenols) are able to modulate the function of the methylation enzymes [119]. Thus, many findings from different studies indicate that the susceptibility of fetal epigenetic modifications can be due to maternal diet, among other factors.

McKay et al. found that unbalanced concentrations of nutrients intake during pregnancy affect the offspring DNA methylation. High levels of vitamin B12 in the maternal blood was correlated with the reduction in the total level of DNA methylation of the neonate, whether the elevated concentration of serum vitamin B12 in the newborn correlated with decrease methylation levels of the insulin-like growth factor-binding protein 3 (IGFBP-3) gene, responsible for intrauterine growth [120]. Low maternal folic acid intake, as well as pregnant women exposed to famine, also affect the expression of insulin growth factor 2 (IGF-2) gene in the offspring by modification of the DNA methylation levels [84]. According to an epigenome-wide analysis, methylated regions that existed in insulin receptors, and carnitine aminotransferase genes that were associated with glucose homeostasis and lipid metabolism, were identified in adults exposed to famine in their fetal life [24,121,122]. Methylation cofactors, such as folate and zinc derived from food, play a role in developing NCD in the offspring. Imbalance of maternal folate concentration affects the development of the fetus [123] through homocysteine buildup, which subsequently affects the fetus organs, including renal and cardiovascular systems [124]. Zinc deficiency in the intrauterine life may have an epigenetic impact on the methylation of promoters, leading to immune pathogenesis [125], increasing the risk of developing CVD and renal disorders [126]. The role of long chain polyunsaturated fatty acids (LC-PUFAs) during gestation was investigated in the offspring health. Khot et al. aimed to look into this aspect and its epigenetic mechanism [127]. The investigators found that low maternal intake of LCPUFAs affects the DNA methylation patterns on angiogenic factor genes, which promotes vascular dysregulation that turns out in rising the cardiovascular risk [127].

The effect of maternal undernutrition on the offspring’s cholesterol dysregulation via epigenetic mechanisms was firstly demonstrated by Sohi et al. in 2011 [128]. This study used rat models that were fed either a low protein diet (LPD), which consisted of 8% protein, or a control diet, consisting of 20% protein. Both diets contained the same amount of Kcal. The authors showed evidence that low protein intake in gestation and lactation period had a role in increasing the cholesterol level in offspring at day 21 after birth. This was found to correlate with the cholesterol 7α-hydroxylase (Cyp7a1) transcriptional repression because of post-translational histone modifications of Cyp7a1 promoter. Cyp7a1 is a rate limiting enzyme that converts the cholesterol into bile acids to regulate the cholesterol level in the body [129]. High circulating cholesterol is one of the risk factors of cardiovascular diseases [130,131]. Additionally, the expression of the Jumonji domain-containing demethylase (*Jmjd2a*) gene in utero decreased as a result of the LPD, which may have contributed to the significant increase in histone H3 trimethylation associated with Cyp7a1 promoter [128]. Similarly, in an experimental rat model, it was shown that maternal protein restriction during pregnancy and lactation led to an impaired glucose tolerance via histone acetylation of the liver X receptor α (LXRα), which regulates the gluconeogenesis in the liver [132]. A similar correlation was also found between maternal protein restriction and histone modifications on the glucose transporter type 4 (GLUT4) gene resulting in the overexpression of GLUT4 in the skeletal muscle of the female offspring [133]^,^ as well as histone modifications on the GATA binding protein 6 (GATA6) gene, which is associated with cardiovascular and metabolic diseases in adulthood [134]. In another animal study, the DNA methylation and histone acetylation in the placenta and liver of the fetus were affected by the abnormal levels of cholesterol and lipid induced by the maternal diet, causing lipid accumulation in the fetal liver [135].

Nutrition also shows an effect on the levels of miRNA, impacting its function in modulating gene expression [136]. Of particular interest in our discussion, specific miRNAs are involved in the regulation of metabolic processes, such as insulin signaling and glucose metabolism, so miRNA dysregulation can be leading to NCDs, for instance T2D [137,138,139] and CVD [137,140]. Moreover, it was shown that maternal malnutrition can impact the miRNA of the fetus, resulting in modified proteins, contributing to the development of obesity and diabetes after birth [141]. Particularly, it was shown that nutrients can influence the expression of miRNAs regulating the folate-mediated one carbon metabolism, which has a role in the regulation of homocysteine, methionine, as well as the protein methylation [142]. Studies on animals reported that both maternal overeating and low protein consumption during pregnancy were associated with significant miRNA dysregulation in the offspring tissues, such as liver and heart [141,143]. These miRNAs are used as biomarkers for some chronic diseases [144,145], which are extracted from various samples, including breast milk [137,146]. An animal study reported that high caloric intake by obese ewes during the peri-conception period led to elevation in expression of specific liver miR-29b, miR-103, and miR-107 in the offspring [147]. In previous reports, miR29b expression was found increased in the liver [148], kidney [148], and pancreatic tissues [149,150] of diabetic animals and human models [151,152]. Moreover, a high level of expressed miR-29b was involved in repressing glucose uptake by insulin via the inhibition of Akt activity, which indicated insulin resistance [153]. An increased serum level of miR-29a was seen in T1D children [154]. The miR-103 and miR-107 are involved in regulation of insulin sensitivity [155] and were found to be upregulated in ewes offspring, signifying that those lambs developed insulin signaling dysregulation [156]. Another form of miR-29 is miR-29c, which targets elastin and collagen in the aorta and functions in maintaining its elasticity. MiR-29c was observed being inhibited in rats of mothers who were undernourished in the gestation period. Thus, the offspring vascular contractility was affected and hypertension was developed [157,158]. Reports on baboons showed that having an obesogenic diet before and after pregnancy promoted significant changes in the cardiac miRNA levels, which are possibly related to cardiovascular diseases [159]. In this study, cardiac miRNAs were assessed and eight of them were differentially expressed; 55 of them were upregulated, and the rest were downregulated in baboons who were born from mothers under high fat/high fructose (HFD) diets compared to those born from mothers who had normal diets during gestation. Some of these miRNAs were also reported in human mapping and contributed to CVD [159].

Maternal vitamin intake was shown to play a key role in the health of the offspring in the long-term; moreover, vitamins were found to have an impact on the expression of miRNAs [160]. A study showed that low vitamin D concentration in pregnant women contributed to their children developing chronic diseases via the expression of miRNAs involved in multiple metabolic pathways [161]. Moreover, low level maternal vitamin B12 was associated with alteration of miRNA expression, which are involved in adipogenesis and insulin metabolism that may initiate metabolic disorders in the offspring [162]. Other micronutrients showing similar effects were polyphenols. These are considered as antioxidants and were found to modify the miRNA expression linked to glucose metabolism, insulin signaling, oxidative stress, and inflammation, which may lead to diabetes and CVD [138,163]. Polyphenol sources are mainly fruits and drinks made from plants [164]. Rat models were used in a recent study to show the harmful effects of maternal consumption of flavonoids from grape seeds in downregulating the offspring hepatic miR-33a, which plays a role in cholesterol regulation, promoting CVD [165]. A study conducted on obese women showed that diet restrictions (for these women to lose weight before pregnancy) led to reprogramming the lipid metabolisms of the offspring, in which the infants’ DNA methylations in liver genes switched to healthy expressions.

#### 4.3.2. Breastfeeding and Epigenetics of NCD

Researchers reported the association between long duration of breastfeeding and reduction in obesity risk later in life [166] and other studies demonstrated the impact of breastfeeding duration on the infants’ DNA methylation [167,168]. According to Obermann-Borst et al., the long period of breastfeeding lowered the CpG methylation of the leptin (*LEP*) gene, which is involved in appetite regulation and fat metabolism, and the *LEP* CpG methylation, was inversely correlated with BMI of children who were at 17 months of age. The reduced methylation of *LEP* caused an increase in the expression and concentration of leptin, a hormone that plays a role in programming the metabolic pathways. In addition, leptin is one of the breast milk constituents; it was proposed that it would likely contribute to programming the neuroendocrine system via methylation of the *LEP* promoter, with a protective effect on childhood obesity [167]. Another study, analyzing the *LEP* gene, found a significant difference between a child’s growth and the *LEP* methylation level. The authors observed higher *LEP* CpG3 methylation when children breastfed for 7 to 9 months and a significant reduction in the weight of the children when they were breastfed for 10 to 12 months [169].

Several studies showed that miRNAs have a crucial function in epigenetic regulation processes as well as intercellular communication [170,171,172]. Moreover, miRNAs are involved in regulation and development of the immune system [173]. Approximately 1400 different miRNAs were identified in breast milk [174,175,176]. Melnik and colleagues found an association between miRNA-148a derived from milk with pancreatic beta-cell differentiation, highlighting that breastfeeding can have a potential protective role against T2D [176]. Human (and some mammals) milk contain abundant levels of miR-125a-5p, which regulate oxysterol-binding protein-related protein (ORP), which has a role in lipid metabolism [177]. It was observed that highly expressed miRNA in human breast milk is involved in production and homeostasis of triglycerides and some were linked to regulation of fatty acid biosynthesis genes [173]. The authors of the aforementioned findings also reported other miRNAs that are involved in regulating lactose synthesis, which occurs in the mammary gland, more specifically, regulating the UDP-glucose transporter, as well as the UDP-galactose transporter [173]. There is very limited evidence on maternal dietary patterns affecting the nutritional content of breast milk [178]. According to a study conducted in nursing rats, it reported that an obesogenic diet may lead to changes in specific miRNA levels in the breast milk. Compared to controls, there was an increase in miR-222 concentration and a decrease in miR-200 and miR-26 in those obesogenic diet-fed rats. The study researchers also mentioned that this type of diet changes the levels of bioactive proteins found in the milk, such as increasing leptin and adiponectin concentrations and decreasing irisin levels, which, consequently, influence the offspring’s metabolism [179]. Another study conducted on humans indicated that maternal nutritional status and diet could affect the expression of miRNA present in breast milk. Findings by Zamanillo et al. showed that leptin, adiponectin, and miRNAs decreased throughout the lactation period in normal weight mothers, while they were altered in the overweight/obese mothers. Moreover, a negative correlation was observed between milk miRNA expression and leptin or adiponectin levels in normal-weight mothers, while there was no correlation observed between those in the overweight/obese mothers. Furthermore, the BMI of infants of normal-weight mothers was negatively correlated, with the miRNAs were miR-103, miR-17, miR-181a, miR-222, miR-let7c, and miR-146b [180].

These findings emphasize the role of epigenetics and nutrition in influencing the metabolism of one’s offspring; the findings also show how these epigenetic modifications are reversible, which can be used in disease treatment and prevention. However, more research needs to be done in humans, as the data scarcity can be due to difficult clinical applications on pregnant women to ensure safety [84].

### 4.4. Nutri-Transcriptomics Role in Offspring NCD

Likewise, the genetic variants and epigenetics are involved in the health outcomes of the offspring due to the nutritional status of the mother; the proteins being translated by those genes can have a crucial role in developing a disease. This concept is called nutri-transcriptomics, as the transcription of the gene depends on the diet, and how nutrients and nutritional status impact the gene expression at the level of mRNA [181].

#### 4.4.1. Maternal Diet and Nutri-Transcriptomics of NCD

A trial study by Al-Garawi et al. investigated the role of vitamin D on gene expression during pregnancy. Thirty pregnant women enrolled in the study were supplemented with vitamin D, and their blood samples were collected in the first and third trimesters. Transcriptional profiles revealed that 5839 genes were significantly differentially expressed, and 14 of them exhibited significant correlation with maternal vitamin D concentrations. Their data suggest that alterations of the gene expression occur during pregnancy and this could be associated with vitamin D supplementation, which contributes toward increasing the concentration of the circulating 25(OH)D precursor [182]. Another study confirming this concept proposed that changes in the gene expression are directly induced by maternal vitamin D intake, which can have an effect on the birth outcome [183]. Other observational studies found that low administration of vitamin D during pregnancy might cause preeclampsia. A study was conducted on female mice models that were divided into two groups, one deficient in vitamin D and the other sufficient with vitamin D; then they were bred with vitamin D-sufficient males. The pregnant mice with low vitamin D concentrations presented high systolic and diastolic blood pressure, and it was continued until 7 days after delivery. Moreover, according to their kidney function analyses, an increase in the mRNA expression for renin and angiotensin II receptors was shown and related to the vitamin deficiency. Surprisingly, when the vitamin D was re-supplemented to the deficient models, it reversed the effect [184]. This was supported by another randomized controlled trial conducted on human pregnant subjects who were supplemented with vitamin D daily throughout their pregnancies. They were classified into two groups, whose subjects with circulating 25(OH)D below 100 nmoles are considered deficient, and above that level were considered sufficient. Their placental mRNAs were analyzed and showed that soluble FMS-like tyrosine kinase 1 (sFlt-1) and vascular endothelial growth factor (VEGF) genes, which have critical functions in the angiogenesis pathway, were significantly downregulated in women who had higher intake of vitamin D (≥100 ng/mL). Thus, maternal vitamin D supplementation may have an impact on the placental gene transcription by downregulation of antiangiogenic factors that can contribute to vascular complications in pregnancy, such as preeclampsia [185]. The relation between an increase of sFlt-1 and preeclampsia was well addressed in many research studies [186,187,188,189]. It was reported that children of preeclamptic mothers were more prone to develop endocrine and metabolic diseases in their first five years of age [190]. In addition, body mass index (BMI) was found higher in males born with preeclamptic mothers compared to those of normotensive pregnancies [191]. Besides this finding, studies on metabolic and inflammatory factors found that the lipid profile differed, and the level of tumor necrosis factor alpha (TNF-α) was higher in the cord blood of children of preeclamptic mothers [192,193,194]. Furthermore, high levels of triglycerides and leptin levels were presented in preeclampsia-exposed children who measured low on the Quantitative Insulin Sensitivity Check Index (QUICKI), which is an assessment used to calculate the insulin resistance degree and the secretory capacity of the pancreatic beta cells [195]. Low levels of QUICKI indicate highly insulin resistance, which is associated with obesity risk and cardiovascular diseases [196].

Animal models were mostly used to study the association between nutrition and transcriptomic background. A significant reduction of hypoxia-inducible factor 1-alpha (HIF-1α) levels was seen in baboon fetuses of HFD-fed mothers, as this factor is known to be crucial for cardiovascular system development [197,198]. Therefore, a reduced level of HIF-1α could precede to abnormal cardiac function during fetal and post-fetal life [159]. In another animal study, two groups of pregnant female mice were assigned to two different diets during conception and lactation periods: one with a normal chow diet (NCD) and the other with an isocaloric low protein diet (LPD). Their findings demonstrated that offspring born from LPD-fed mothers exhibited LBW, glucose intolerance, and a decrease in insulin secretion. The same animals also expressed some of miRNAs at weaning stage, and all of these expressed miRNAs were involved in inflammatory pathways with a high levels of serum pro-inflammatory IL-6 and TNF-α cytokines. In addition, the mRNA and protein expression of these cytokines was a significant high level in those offspring. Hence, these results illustrate that consuming low protein levels during pregnancy affects miRNA expression, which might be linked to chronic inflammation and glucose intolerance in offspring [199].

A study on pigs focused on nutrient intake during pregnancy and its influence on the offspring developing metabolic problems. Cai et al. observed that pig mothers who consumed betaine while they were pregnant gave birth to newborn pigs who presented elevated serum and hepatic betaine contents, with a significant upregulation of the hepatic enzyme glycine *N*-methyltransferase (GNMT). Moreover, liver cholesterol was higher and the expression of the cholesterol metabolic genes was altered in the neonate piglets. Cholesterol homeostasis is maintained through certain factors, such as 3-hydroxy-3-methylglutaryl CoA reductase (HMGCR) and sterol regulatory element-binding protein-2 (SREBP2) were both downregulated at mRNA level. The enzymes cholesterol-27α-hydroxylase (CYP27α1) and cholesterol-7α-hydroxylase (CYP7α1), which function in bile acid transformation, were upregulated at the mRNA levels [200]. Moreover, the same author conducted another study on the role of betaine in galactose metabolism of piglets. The study reported that neonate piglets exposed to betaine in their fetal lives showed low concentrations of galactose in their blood, which was correlated with a significant downregulation of the galactokinase-1 (GALK1) hepatic gene, involved in galactose breakdown process. This enzyme deficiency was associated with increase galactose in the newborn’s blood, as galactose can pass freely from the mother through the placenta to the fetus, and its elevated level causes liver toxicity. Thus, as betaine is involved in galactose and cholesterol homeostasis, and its deficiency could be associated with lipid disorder and diabetes, this study suggests that betaine can play an effective role in the offspring’s health via genetics, transcriptomics, and epigenetics [201].

#### 4.4.2. Breastfeeding and Nutri-Transcriptomics of NCD

Few research studies have investigated the effects of breastfeeding in nutri-transcriptomics and its role in NCD development. Cheshmeh et al., in a case-control study involving 150 infants, aged between 5 and 6 months, investigated the impact of breastfeeding and formula feeding on the expression of obesity-related genes, which are fat mass and obesity-associated (*FTO*), carnitine palmitoyltransferase 1A (*CPT1A*), and peroxisome proliferator-activated receptor-α (*PPAR-α*). The subjects were categorized into three groups: breastfed only, formula-fed only, and fed with both milks. The study findings showed that infants fed formula milk in either groups exhibited higher weight as well as higher expression of *FTO* and *CPT1A* genes, and a lower expression of *PPAR*-α gene when compared to the exclusive breastfeeding group. The authors concluded that breastfeeding apparently showed a protective effect against obesity, modulating the expression of obesity-related genes [202]. A parallel, multicentric study measured the expression levels of *PPARα* and *CPT1A* genes, together with other genes, such as solute carrier family 27 member 2 (*SLC27A2*), fatty acid synthase (*FASN*), insulin receptor (*INSR*), and leptin receptor (*LEPR*), which are used as transcriptional biomarkers of the metabolic status in children from 2 to 9 years old from eight different European countries. The children who were breastfed showed higher expressions of *SLC27A2*, *FASN*, *PPARα,* and *INSR*, and were at lower risk to develop obesity. On the other hand, an increase in triglycerides levels was shown in formula-fed children who also had low expression of these genes. According to these data, higher expressions of *SLC27A2*, *FASN*, *PPARα,* and *INSR* genes in the children’s blood reflected a protective role of breastfeeding, as these genes are indicators of a lower risk of developing insulin resistance and dyslipidemia linked with obesity in children. In addition, these biomarkers are likely to distinguish—among formula-fed children—the ones that are at a high risk of metabolic changes [203].

Thus, transcriptomics is very helpful in terms of identifying the genes that express RNAs that have been modulated by diet, specifically nutrients taken during pregnancy and breastfeeding, and then correlate the physiological and pathological long-term effects on the offspring. The main studies are summarized in Figure 1.

## 5. Conclusions

Maternal diet and nutritional status, as well as single nutrient intake during the entire course of pregnancy and breastfeeding, were demonstrated to affect fetal molecular pathways, such as lipid and liver profiles, inflammation, and angiogenesis, which contribute to the development of NCDs in childhood and adult life. These pathways are regulated at multiple levels via the gut microbiota, through epigenetic modifications, ending with affecting gene expression and protein function (Figure 2). The application of omics technologies to study the role of maternal diet and breastfeeding on the offspring’s health status is able to detect the modifications at a molecular level. However, more studies are needed to confirm the current findings; moreover, it is recommended to move to an integrated multi-omics approach that is able to dissect the multiple interactions among nutrients, microbiota metabolites, and the mutual effects on genes modifications and expression. Moreover, population-specific and patient-personalized studies will contribute to the development of a personalized nutrition approach to prevent and treat NCD from the fetal life stage.

## Figures and Tables

**Figure 1 life-11-00467-f001:**
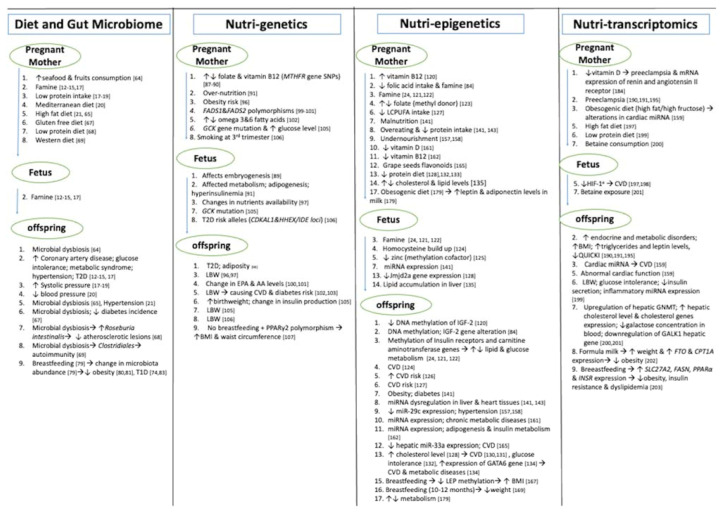
The effect of mother’s nutrition during pregnancy on the fetus and offspring health has been associated with microbiome, genetics, epigenetics, and transcriptomics in humans and non-human models. A repeated number in the lists indicates the pathway to follow from the mother to fetus and to offspring in each omics separately. Numbers not indicated show that there is no stated association. Arrows: (↑) represents the increase, (↓) represents the decrease, (↑↓) represents the disturbance/imbalance, (→) represents the leading cause.

**Figure 2 life-11-00467-f002:**
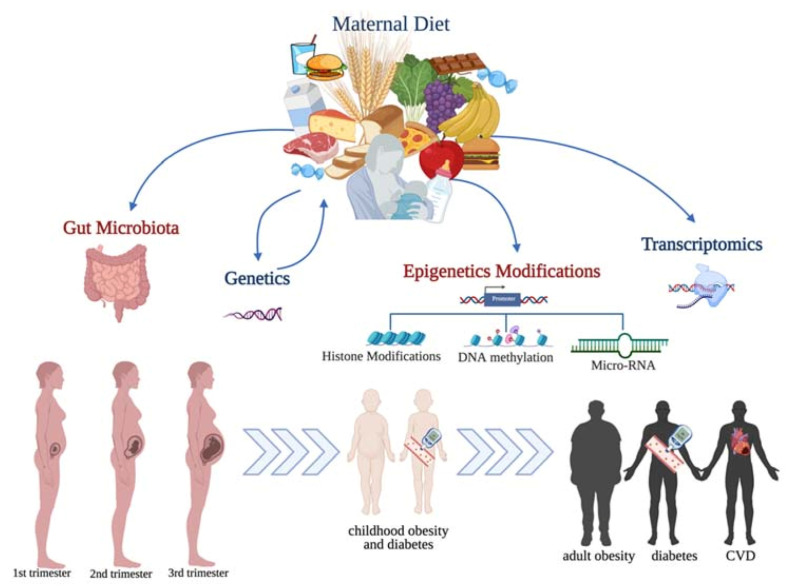
Precision nutrition applied to pregnancy to understand the mechanisms that lead to NCD in childhood and adult life. Created with BioRender.com (accessed on 7 May 2021).

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
