# Peer review of "Personalized Nutrition Approach in Pregnancy and Early Life to Tackle Childhood and Adult Non-Communicable Diseases"

_life, 2021, doi:10.3390/life11060467_

Round 1
Reviewer 1 Report
This is an exhaustive and comprehensive review on the personalized approach to nutrition of pregnant and lactating women to address non-communicable pathologies in offspring.
The only comments I have are to change in page 4 line 2: placental amniotic fluid, remove placental
Page 4 line 3: mother placenta, remove mother. The placenta belongs to the fetus
Author Response
We would like to thank the reviewers for their efforts, the comments and suggestions that they provided to improve this review paper. Below you can find our responses to each comment given by the reviewers.
All the additional or edited parts as per reviewers’ suggestions were colored in red within the manuscript.
Comments of Reviewer #1:
Comment: to change in page 4 line 2: placental amniotic fluid, remove placental.
Response: the word placental has been removed.
Comment: mother placenta, remove mother. The placenta belongs to the fetus.
Response: the word mother has been removed.

Reviewer 2 Report
The manuscript entitled: Personalized nutrition approach in pregnancy and early life to tackle childhood and adult non-communicable diseases, by Al-Abduljabbar et al., discusses the most recent evidence on the role of maternal nutrition in the development of NCD using omics technologies to determine the molecular mechanisms. The topic is novel and of interest to researchers as it gives a nice overview. The authors conducted a comprehensive literature review. However, the section could be better organized with subsections to improve the readability. Few comments are listed below to improve this manuscript.
- The manuscript needs the grammatical edition.
- Abbreviation must be checked.
- Avoid using ‘they’ in manuscript (specify the authors/research team)
- Please add a definition for non-communicable disease (NCD) in the first paragraph of Introduction (there is a few examples but better to add a definition)
- Please explain more about “window of susceptibility”
- Add section in introduction: on early life nutrition (breast vs formula) in the determining the development of NCDs later in life
- Maybe better to add definition of epigenetic, metabolomics and microbiome and … in related paragraphs
- 1 Diet and gut microbiota in pregnancy and role in offspring NCD Please organize the paragraphs better: eg microbiome and pregnancy (and risk of NCD (Obesity, diabetes) and then breastfeeding vs formula.
- It is suggested to classify each section based on each NCD for more organization and easy follow: for example: in the section “2 Nutrigenetics in pregnancy and role in offspring NCD” you can add subtitles: Obesity, Diabetes…
- 3 Nutri-epigenomics in pregnancy and role in offspring NCD
- Should define the types of epigenetic changes.
- Add subsection (or different paragraphs for each idea or topic).
- 4 Nutri-transcriptomics in pregnancy and role in offspring NCD- This is only one paragraph should be separated in multiple thematic paragraph such as Vitamin D, obesogenic diet, high fat/high fructose (HFD) diet, isocaloric low protein diet, betaine, etc..
- Table 1: numbering of items is sometimes wrong and sometimes twice the same idea (eg 5, and 6 in microbiome/pregnant mother)
Author Response
Comment: Abbreviation must be checked.
Response: the abbreviations used in the manuscript have been checked, and the full name of the (HHEX/IDE) genes is added within the text.
Comment: Avoid using ‘they’ in manuscript (specify the authors/research team).
Response: all “They” which refers to the authors have been deleted and replaced.
Comment: Please add a definition for non-communicable disease (NCD) in the first paragraph of Introduction (there is a few examples but better to add a definition)
Response: the definition of the non-communicable disease (NCD) has been added in the introduction section.
Comment: Please explain more about “window of susceptibility”.
Response: more explanation of “Window of susceptibility” is added in the introduction section.
Comment: Add section in introduction: on early life nutrition (breast vs formula) in the determining the development of NCDs later in life
Response: a new section about Breastfeeding & Formula-feeding in determining the development of NCDs later in life is added in the introduction, it is numbered as section 3.
Comment: Maybe better to add definition of epigenetic, metabolomics and microbiome and … in related paragraphs
Response: the definitions of microbiome, nutrigenetics, and transcriptomics are newly added in the related paragraphs. The definition of epigenetics had been mentioned in the related section earlier.
Comments:
- 1 Diet and gut microbiota in pregnancy and role in offspring NCD; Please organize the paragraphs better: eg microbiome and pregnancy (and risk of NCD (Obesity, diabetes) and then breastfeeding vs formula.
- It is suggested to classify each section based on each NCD for more organization and easy follow: for example: in the section “2 Nutrigenetics in pregnancy and role in offspring NCD” you can add subtitles: Obesity, Diabetes.
- 4 Nutri-transcriptomics in pregnancy and role in offspring NCD- This is only one paragraph should be separated in multiple thematic paragraph such as Vitamin D, obesogenic diet, high fat/high fructose (HFD) diet, isocaloric low protein diet, betaine, etc..
Response: thank you for your suggestions and comments. We agree on reorganizing the contents. Accordingly, we divided each omics section into two subsections: maternal diet and breastfeeding. Breastfeeding sections have been newly added into epigenetics and transcriptomics sections.
Comment: 3 Nutri-epigenomics in pregnancy and role in offspring NCD; Should define the types of epigenetic changes and Add subsection (or different paragraphs for each idea or topic).
Response: the definitions of epigenetic changes or mechanisms are added in the Nutri-epigenomics in pregnancy and role in offspring NCD section. Each idea or topic of the epigenetic changes were divided into separate paragraphs. Histone modification mechanism has been newly added into the section.
Comment: Table 1: numbering of items is sometimes wrong and sometimes twice the same idea (eg 5, and 6 in microbiome/pregnant mother).
Response: the numbering in the table is checked and edited. The repetitions of the ideas were merged in one idea/ number. New information is added in the table according to new additional information in the manuscript.

Round 2
Reviewer 2 Report
The manuscript has been revised appropriately. However, some English language corrections are required.